# Myopia-correcting lenses decrease eye fatigue in a visual search task for both adolescents and adults

Hyeongsuk Ryu[1], Uijong Ju[2], Christian Wallraven[1,3]*

1 Department of Brain and Cognitive Engineering, Korea University, Seoul, South Korea, 2 Department of Information Display, Kyunghee University, Seoul, South Korea, 3 Department of Artificial Intelligence, Korea University, Seoul, South Korea

* wallraven@korea.ac.kr

## Abstract

The steady, world-wide increase in myopia prevalence in children over the past decades has raised concerns. As an early intervention for axial-length-related myopia, correcting lenses have been developed (such as Defocus Incorporated Multiple Segment (DIMS) lenses), which have been shown to be effective in slowing myopia progression. Beyond this direct effect, however, it is not known whether such lenses also affect other aspects important to the wearer, such as eye fatigue, and how such effects may differ across age, as these lenses so far are typically only tested with adolescents. In the present work, we therefore investigated perceived fatigue levels according to lens type (normal vs DIMS) and age (adolescents vs adults) in a demanding visual search task ("Finding Wally") at two difficulty levels (easy vs difficult). Whereas age and difficulty did not result in significant differences in eye fatigue, we found a clear reduction of fatigue levels in both age groups when wearing the correcting lenses. Hence, the additional accommodation of these lens types may result in less strain in a task requiring sustained eye movements at near viewing distances.

## Introduction

It is estimated that by 2050 the global myopia prevalence will be around 5 billion people [1]. Although a non-life-threatening disorder, myopia has far-reaching consequences in the health sector resulting in significant social and economic spending on both personal and national levels [2]. One of the major optical sources of myopia stems from an excessive elongation of the eye's axial length that happens during childhood growth [1, 3, 4], leading to decreased visual acuity at far viewing distances. Connected with this, other researchers have argued [5–8] that societies' increase in near-work activities and an accompanying indoor lifestyle is associated with increased risk of myopia—a trend that has only been exacerbated in the COVID-19 pandemic with extended lockdowns, home-schooling, reduced outdoor activities, and the ever-increasing use of electronic devices for communication.

Several clinical interventions have been developed with the aim of preventing or slowing the progression of myopia in children; broadly, these can be subdivided into topical

**Data Availability Statement:** Behavioral data is available at https://osf.io/hdmt2/?view_only=8e7fefec11eb4716b9cecf9a5c0e3435.

**Funding:** This work was supported by the Institute of Information & Communications Technology

Planning & evaluation (IITP) grant funded by the Korean government (MSIT) (No. 2019-0-00079, Artificial Intelligence Graduate School Program (Korea University)). Normal single vision lenses and DIMS lenses were donated by the Korean office of the HOYA Corporation, Japan. The HOYA corporation had no influence on the study design, analysis, or reporting of the results.

**Competing interests:** The authors have declared that no competing interests exist.

pharmaceutical agents or optical interventions using contact lenses or specially-treated glasses [9]. Pharmaceutical treatments have been deemed effective; however, they also suffer from side effects of long-term use, including photophobia, glare, and accommodation loss [10–12]. Similarly, defocus incorporated soft contact (DISC) lenses can be effective [13]. The basic mechanism of these lenses is based on monocular deprivation (MD), which modifies the ocular balance resulting in a boost to the deprived eye [14]. In imaging studies, it has been shown that even a short MD reduced GABA concentrations in early visual cortex and that this reduction was correlated with the deprived eye's perceptual boost as measured through binocular rivalry [15]. While effective, such contact lenses at the same time require careful management for eye health as shown in [16], who demonstrated increased risks of infection from contact lenses due to inappropriate lens handling or cleaning. Another optical intervention uses defocus incorporated multiple segments (DIMS) lenses, which are a form of spectacle lenses based on a similar defocus mechanism as DISC [13]. In a recent clinical trial following young children for two years, DIMS lenses significantly slowed myopia progression compared to single vision (SV) lenses [17]. Such lenses do not share the side effects of pharmaceutical treatments, nor do they require special care as contact lenses do.

Whereas clinical trials have typically focused on the assessment of the main outcome variable of myopia progression, a factor that has received comparatively little attention is that of eye fatigue or eye strain (asthenopia). Indeed, as is well known, changing the refractive power in prescriptions for people with myopia or astigmatism is often accompanied by symptoms such as asthenopia, headache, or dizziness. Up to four weeks of adaptation may be necessary for these symptoms to subside [18]. This adaptation process is similar not only for adults but also for children [19]. Given that the DIMS lenses change the refractive power of a subpart of the visual field, it is therefore important to track such differences in wearing comfort.

Overall, asthenopia research in adults and children indicated correlations of eye strain with age, refractive error, and insufficient accommodation [20–22]—a convex lens, however, was shown to not only help accommodation, but also to support orthoptic exercises for near distance tasks [23, 24] and therefore was able to reduce eye strain [25, 26]. Similarly, as short-wavelength light (380-420nm) is a critical factor in inducing asthenopia [27–29], a lens surface coated to block these wavelengths can decrease eye fatigue, coupled even with increases in visual performance and sleep quality [30, 31]. Lin et al. [32] investigated the effect of such lenses using critical flicker fusion (CFF) and showed reduced eye strain as well. As the DIMS lenses change the relative composition of focal and peripheral parts of the visual field, it is important to also investigate whether they would lead to changes in eye strain. A recent study on DIMS lenses with two cohorts of children and adults traced wearing comfort in both groups after one week, comparing single vision (SV) lenses with DIMS lenses [33] with a one-time survey. The authors found no clear differences in the children group on measures including eye strain, nausea, or dizziness, but showed that adults tended to feel more nausea and dizziness for the DIMS compared to the SV lenses. For both groups, overall acceptance of the DIMS lenses given their potential for myopia control after the initial acclimatization phase was high.

The aforementioned study focused on evaluating the wearing comfort of myopia-control lenses with a one-time subjective evaluation. Our main aim in this study, however, was to investigate wearing comfort of such lenses for this first time in a more fine-grained manner *during* a demanding visual task, hence gathering more data on how SV and DIMS lenses may differentially impact perceived eye strain. For this purpose, the study by Lin et al. [32] demonstrates an interesting approach, in which a well-researched perceptual task or paradigm (like flicker fusion) is employed as a vehicle for studying factors related to eye strain and visual performance. In this manuscript, we take a similar route, using a paradigm taken from the visual

search literature to investigate potential effects of DIMS lenses. Specifically, here we focus on the "Where's Waldo" (also called "Finding Wally") visual search puzzles. Visual search is a core perceptual task, involving the search for a particular target among a (usually complex) background [34, 35]. The "Where's Waldo" task itself is one example of a visual search task, which has been used in several previous studies: Port et al. [36], for example, used the task to measure saccade properties across the lifespan and Sahraian et al. [37] showed that training with this task also improved target detection in radiological images. In this context, Casco et al. [38] showed that visual search performance critically depends on age, and, furthermore, work by Solimini et al. [39] indicates that eye fatigue during game play also is different by age. Given the additional, wide-ranging differences in developing and adult eyes [40] and the potential differences in sensitivity to the DIMS lenses in adults and children [33], it will therefore be important to also test how such lenses may affect adults versus adolescents differently.

In summary, this study aims to determine whether DIMS lenses (compared to SV lenses) impact eye fatigue and visual performance in a demanding visual search task, comparing two different age cohorts.

## Materials and methods

### Participants

We recruited 20 adults and 22 adolescents with myopia. Adults were recruited from Korea University (females: 10; mean age: 24.85years, SD: 3.68), whereas adolescents were recruited from two middle schools (Chungju Buk Girls Middle School and Kamgok Boys Middle School; females: 12, mean age: 14.64years, SD: 0.58). In our selection of participants, people were excluded (1) if they wore prismatic glasses (due to an officially diagnosed strabismus), (2) if vision correction above the default dispensing range of DIMS lenses (spherical: < -10Diopter, cylindrical: <-4.00Diopter) would be required. The myopia ranges for both groups were similar (Adults: Right Eye_SE (Spherical Equivalent): -3.59, SD: 2.37, Left Eye_SE (Spherical Equivalent): -3.21, SD: 2.38; Adolescents: Right Eye_SE: -3.69, SD: 2.10, Left Eye_SE: -3.57, SD: 1.75; Right Eye: $t(38) = 0.15$, $p = .880$, Left Eye: $t(35) = 0.56$, $p = .582$). The study followed the tenets of the Declaration of Helsinki, and written informed consent was obtained from guardians of the middle school students and from adult participants after explanation of the nature and possible consequences of the study. The study and research protocols were approved by the Internal Review Board (IRB) of Korea University (KUIRB-2019-0310-01).

### Lenses

We used commercially-available coated clear standard SV and DIMS lenses manufactured by HOYA (https://www.hoyavision.com) and fitted to each participant's myopia. Fig 1 shows the glasses used for the experiment and their effect on an example picture; unlike general SV lenses, the DIMS lens has a corrected, clear view only in a 9.4mm limited optical zone in the middle of the glasses, which is surrounded by a 33mm defocus area containing a honey-comb pattern of +3.50 diopter reading spheres. The different optical zones result in a blurred region subtending roughly 69˚ x 57˚ visual angle (horizontal x vertical) that excludes a central 13˚ non-blurred region.

### Stimuli

"Where is Wally?" (also known as "Waldo" in other countries) is a visual search puzzle introduced by the British author Martin Handford in 1987. A series of children's illustration books shows detailed drawings of a crowd. The aim of the search task is to find the bespectacled

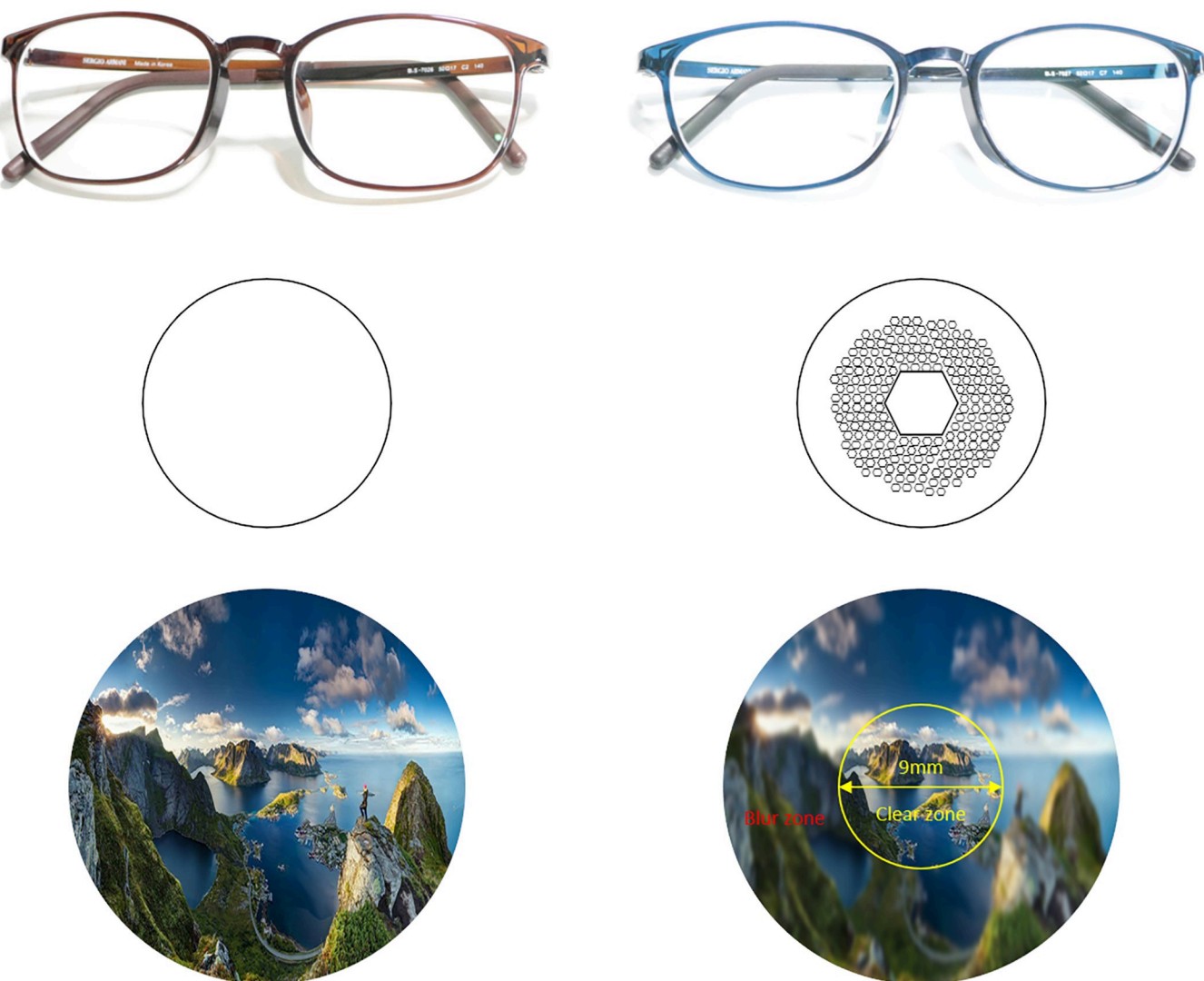

**Fig 1. Single vision (SV) lenses on the left and defocus incorporated multiple segments (DIMS) on the right.** The visual effect of the DIMS design is shown as well for an example picture.

Wally, who wears a red and-white striped shirt, blue pants, bobble hat and is hidden somewhere on the page among the crowd.

A first sample of 24 pictures was obtained from the "Where is Wally?" books. In order to make the search task better to analyze, we subdivided each of the images into a regular 7x7 grid indexed by letters ('a—g') and numbers ('1–7'). The 7x7 grid was overlaid on the stimulus images and shown during all experiments. The task for participants was to view the image and to indicate which of the target grid squares contained Wally. Based on initial inspection, we grouped the images into two different difficulty levels (Easy and Difficult) and showed these to a group of N = 10 pilot participants for validation. The number of distinct human characters in one grid square was adopted as a reference for the difficulty level decision (Easy: Min_average: 3.1ea (SD = 2.03), Max_average: 9.3ea (SD = 2.00), an example of this category is the image entitled 'FVN and games in ancient Rome'. Difficult: Min_average: 6.6ea (SD = 3.89), Max_average: 15.5ea (SD = 6.79), an example of this category is the image entitled 'The Mighty

Fruit Fight'–the difference in this metric was significant for both minimum and maximum values (paired t-test, Min: t(18) = -2.52, p = .021, Max: t(18) = -2.77, p = .013)).

The Easy pictures on average received a correct response in 37.29sec (SD = 7.78), whereas the Difficult images received a correct response in 152.09sec (SD = 62.42) on average—this difference was significant (paired t-test, t(9) = 6.12, p < .001). Of the 24 images, four images that resulted in extremely long average search times were excluded, resulting in 20 final images. These images were grouped into four sets with each set containing both Easy and Difficult images–each set had similar response times on average (Set 1: 88.13sec (SD = 50.62), Set 2: 93.68sec (SD = 54.82), Set 3: 91.66sec (SD = 40.51), Set 4: 96.67sec (SD = 39.95)). Images were randomized within each set and sets were randomized across participants.

In all pilot and main experiments, stimuli were generated and controlled using MATLAB (Mathworks, Natick, USA, 2019a) and Psychtoolbox-3 (3.0.16) for Windows, running on a laptop computer (XPS 15 9550, Dell, USA). Stimuli were shown on a 15.6" FHD (1920X1080) monitor at a comfortable viewing distance of 60 cm.

In the context of measuring eye fatigue, self-report questionnaires, critical flicker fusion frequency (CFF) [32], and blink rate measurements [41] have mainly been used as measures. Due to regulatory limitations in the ongoing COVID-19 pandemic, we were not able to employ eye-tracking with our participants, so that we decided to focus on self-report-based questionnaires in the present study [23, 39, 42]. This fatigue assessment was done across five levels, each of which was explained to participants beforehand: 1. None to very little: You did not feel any strain when finding Wally.; 2. A little: You felt a little tired as you searched for Wally; 3. Moderate: You felt tired as you searched for Wally, but it can be sustain; 4. Considerable: You feel stressed from the search task, your eyes or the area around it may feel itchy (especially the canthus), you may have had to yawn from time to time.; 5. Very Much: You found it very difficult to concentrate on the search task as your eyes felt very tired, itching or even pressure in the eyes may have occurred, you may have needed to yawn. The questionnaire was designed based on similar layouts used in previous studies [43].

## Procedure

All participants first were invited for a standard optometric test to determine the best fitting lens parameters for the DIMS lenses. Upon arrival of the lenses, these were fitted to a frame and then handed over to the participants for a period of at least two weeks (Adults: 15.45days (SD = 5.42), Adolescents: 16.41day (SD = 3.05)). We confirmed whether participants adapted to the DIMS lenses through self-report by calling them after 1, 3, 5, 7, and 14 days and inquiring about any wearing discomfort (such as dizziness or headache)—participants responded that discomfort disappeared after a maximum of 7 days (Adults: 6.45 days (SD = 2.89), Youths: 5.23 days (SD = 2.14))–a period after which the main experiment then started.

Before the main experiment, participants were first acquainted with the setup and were informed about that we wanted to measure the effect of DIMS versus SV lenses with performance in a visual search task. Specifically, we explained that after each picture appeared on the monitor, they were to search for Wally and to indicate the letter and digit of the figure as soon as they had found it using a mouse button press on the target square. After the picture appeared, their answer was recorded as correct if the participant clicked in the square containing the Wally character. The time taken to press the button after stimulus onset was recorded as 'Response Time (RT)'. Half of the participants in either Adolescents and Adults groups group watched the first two sets with SV lenses and then switched to DIMS lenses, whereas the other half did the opposite. We made sure that participants were easily able to switch the lenses without experiencing discomfort.

After every stimulus, participants were also asked to rate their subjective eye strain using the self-report questionnaire although care was taken not to stress that this measure was the main measure of interest in this experiment. After each set, participants were able to take a 2-minute break.

### Data analysis

Dependent variables consisted of accuracy, response time, and perceived eye fatigue and were analyzed for factors of age group (Adult, Adolescents), lens type (SV, DIMS), and difficulty (Easy, Difficult—as defined in pilot experiment). Prior to the main testing, Shapiro-Wilk tests and additional Skewness and Kurtosis checks were conducted to confirm normality, followed by transformations if necessary. Mixed ANOVAs were used for analyses of main effects and interactions using SPSS (Version 25.0, SPSS Inc., Chicago, IL, USA).

## Results

Average values and standard deviations for the three dependent variables are shown in Table 1 as a function of age group and lens type. Given the large number of trials (n = 840), Shapiro-Wilk tests indicated violation of normality for all variables, however, skewness and kurtosis were within ±2 for Accuracy and Eye Fatigue. The variable of Response Time showed a highly skewed, left-tailed histogram, and therefore was log-transformed before analysis with the mixed ANOVA.

### Accuracy

Accuracy showed a significant main effect of age group ($F(1,40) = 5.63$, $p = .023$, partial $\eta^2 = .64$) with adults having an average of 10% higher accuracy (see Table 1). No other effects were significant (Table 2).

### Response time

In addition to adults being more accurate, their response time was also slightly faster on average ($F(1,40) = 4.78$, $p = .04$, partial $\eta^2 = .57$). Similarly, and as expected from the pilot results, response time also showed a main effect of difficulty ($F(1,40) = 11.54$, $p < .01$, partial $\eta^2 = .91$) (Table 3) with Easy images (mean = 69.88s) being on average around 18s faster to solve than Difficult images (mean = 88.31s). There were no other effects on response time.

### Eye fatigue

Results for eye fatigue–our main measure of interest–showed main effects of lens type ($F(1,40) = 32.10$, $p < .001$, partial $\eta^2 = 1.0$) and difficulty ($F(1,40) = 8.22$, $p = .007$, partial $\eta^2 = .80$) (Table 4, Fig 2).

Starting with difficulty, eye fatigue levels were—not surprisingly—higher for Difficult (mean = 2.70) than for Easy (mean = 2.53) images. Results for the main effect of lens type

**Table 1. Mean and SD values for accuracy, Response Time (RT), and eye fatigue across adults and adolescents for the two types of lenses (SV and DIMS).**

|  |  | Accuracy (SD) | RT (SD) | Eye Fatigue (SD) |
|---|---|---|---|---|
| Adults | SV | 0.77 (0.16) | 63.22 (40.59) | 2.72 (0.58) |
|  | DIMS | 0.83 (0.13) | 63.92 (37.04) | 2.22 (0.70) |
| Adolescents | SV | 0.70 (0.20) | 80.63 (35.20) | 3.17 (0.65) |
|  | DIMS | 0.70 (0.16) | 73.75 (30.74) | 2.35 (0.67) |

**Table 2. Mixed ANOVA results for the dependent variable of accuracy.**

|  | Mean Square | F | Sig | Partial η² |
|---|---|---|---|---|
| Group | 0.44 | 5.63 | 0.02* [a] | 0.64 |
| Lens | 0.04 | 1.38 | .25 | 0.21 |
| Lens*Group | 0.04 | 1.38 | .25 | 0.21 |
| Difficulty | 0.12 | 2.83 | .10 | 0.38 |
| Difficulty*Group | 0.03 | 0.80 | .38 | 0.14 |
| Lens*Difficulty | 0.01 | 0.40 | .53 | 0.10 |
| Lens*Difficulty*Group | 0.02 | 1.16 | .29 | 0.18 |

[a] *p < .05.

**Table 3. Mixed ANOVA results for the dependent variable of log-transformed response time.**

|  | Mean Square | F | Sig | Partial η² |
|---|---|---|---|---|
| Group | 0.57 | 4.78 | .04 | 0.57 |
| Lens | 0.004 | 0.11 | .75 | 0.10 |
| Lens*Group | 0.004 | 0.10 | .76 | 0.10 |
| Difficulty | 0.28 | 11.54 | < .01** [a] | 0.91 |
| Difficulty*Group | 0.004 | 0.16 | .69 | 0.07 |
| Lens*Difficulty | 0.03 | 1.29 | .26 | 0.20 |
| Lens*Difficulty*Group | 0.02 | 0.97 | .33 | 0.16 |

[a] **p < .01.

**Table 4. Mixed ANOVA results for the dependent variable of eye fatigue.**

|  | Mean Square | F | Sig | Partial η² |
|---|---|---|---|---|
| Group | 3.45 | 3.09 | .09 | 0.40 |
| Lens | 18.33 | 32.10 | < .001*** [a] | 1.0 |
| Lens*Group | 1.09 | 1.91 | .18 | 0.27 |
| Difficulty | 1.30 | 8.22 | .01** [b] | 0.80 |
| Difficulty*Group | 0.12 | 0.76 | .39 | 0.14 |
| Lens*Difficulty | 0.06 | 0.57 | .46 | 0.11 |
| Lens*Difficulty*Group | 0.10 | 0.91 | .35 | 0.15 |

[a] ***p < .001.

[b] **p < .01.

showed that DIMS lenses significantly reduced eye fatigue on average by 23% (SV: mean = 2.94; DIMS: mean = 2.28). We did not find evidence for an interaction, and Fig 2 shows that both groups' fatigue levels were reduced considerably by the DIMS lenses (for SV lenses: Adolescents: 2.72 vs Adults: 3.17) to similar levels for the DIMS lenses (for DIMS lenses: Adolescents: 2.22 vs Adults: 2.35).

## Effect of trials

Finally, we take a look at how the measures of Response Time and Eye Fatigue developed over the course of the experiment. For this, we plot both measures as a function of trial (1 to 10)

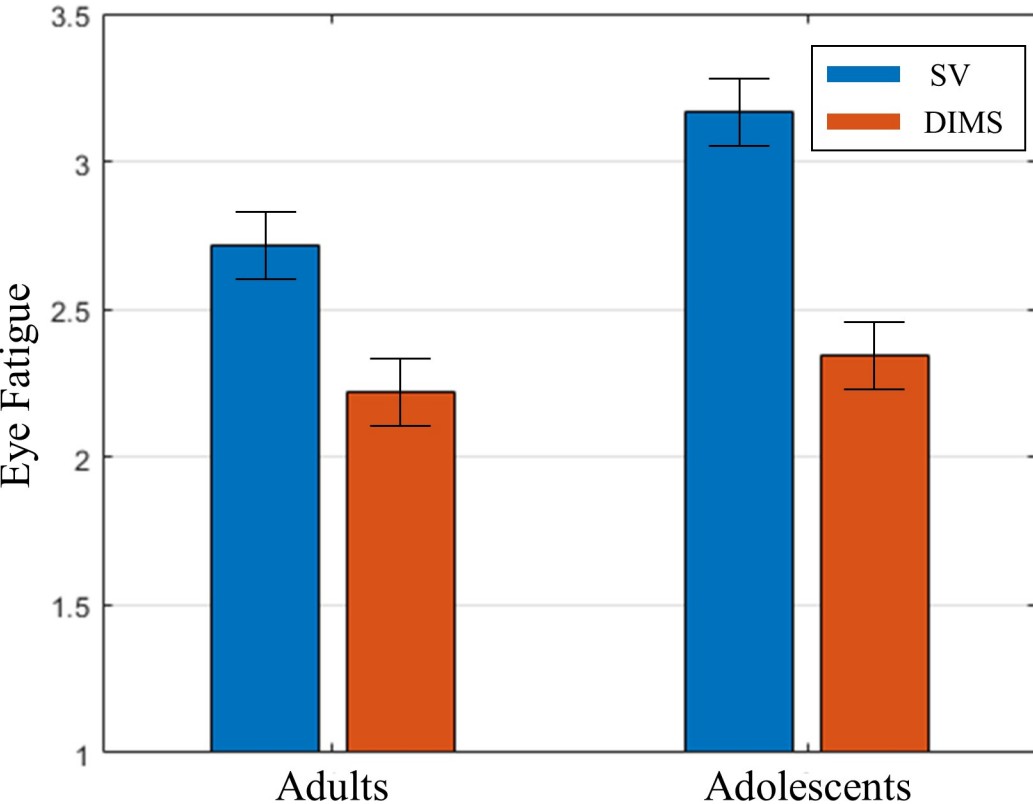

**Fig 2. Results for eye fatigue ratings as a function of group and lens type.** Although the Adolescent group seemed to have slightly higher ratings in the SV lenses compared to Adults, the interaction did not reach significance.

and lens type (SV vs DIMS) in Fig 3. Both measures show a slight upward trend for both DIMS and SV lenses as the number of trials increases, indicating increasing fatigue levels. The two measures are also correlated significantly, as indicated by a Spearman correlation value of r = .377 (p < .001), showing that increased eye fatigue also goes along with increased response

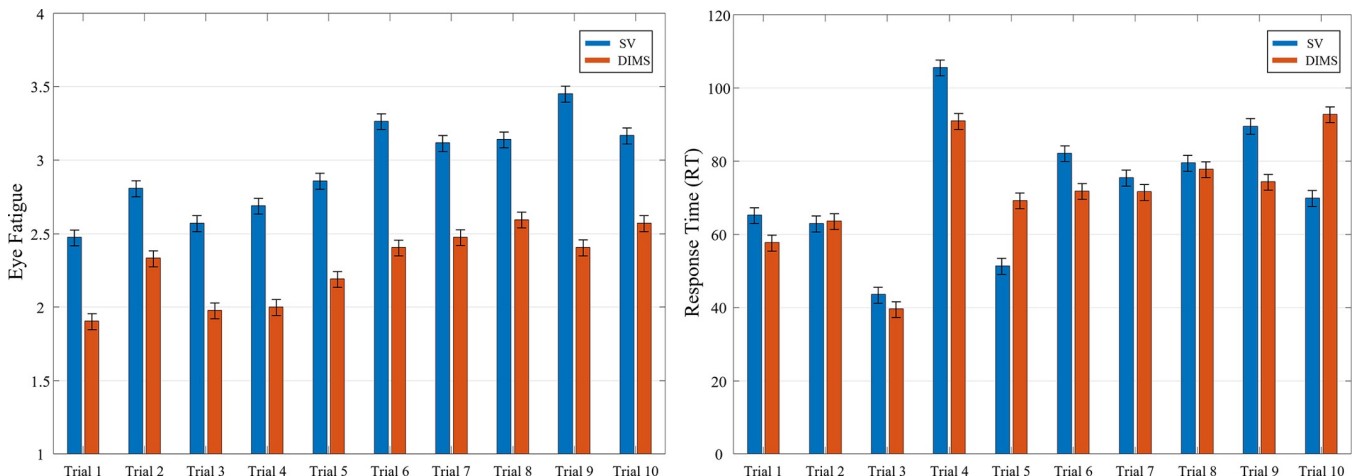

**Fig 3.** a) eye fatigue ratings and b) response times as a function of trial numbers across the experiment for the SV and DIMS lens conditions. The shaded areas indicate the confidence interval of the mean across participants.

times. Importantly, however, levels for DIMS lenses stay continuously lower than those for SV lenses across all trials (Fig 3A), whereas response times are similar across lens types (Fig 3B).

## Discussion

The purpose of this study was to investigate how DIMS lenses—originally designed to help with axial myopia—would affect self-rated eye strain levels during a difficult search task compared to standard lenses. The study used a popular visual search paradigm ("Where's Waldo") in which a target figure was to be found in a highly-cluttered visual environment containing lots of distractor objects. In addition, two different age groups with similar, initial myopia levels were tested to see whether potential fatigue effects of the lenses would be restricted to the developing eye only.

Participants in our study reported a considerable decrease in eye fatigue levels while wearing the DIMS lenses to which they had been accustomed before. This effect was observed for both the young (Adolescents) and older (Adults) group, indicating that the strain from the search task was reduced regardless of age for DIMS lenses. This is especially noteworthy as a previous study in which a one-time survey was employed to investigate wearing comfort found that adults experienced more discomfort when wearing DIMS lenses compared to SV lenses–an effect that could also have been driven by the relatively smaller sample size in that study. In contrast, in the context of our *continuous* evaluation of eye strain, we found that both groups of participants showed a clear reduction in strain of DIMS compared to SV lenses.

One potential explanation for this result may be related to changes in accommodation stemming from the additional optical power of the lens: as is known, small dioptric changes in the so-called accommodative micro-fluctuations (AMF) during the accommodation process results in continued eye stress. This was used by Kajita et al. [44], who compared spherical lenses with aspherical lenses via AMF variation during screen reading and confirmed that the aspheric lens design reduced ciliary muscle stress. Hence, the lens's design was able to diminish regulatory muscle activity, leading in turn to reduced eye tiredness. Similarly, multifocal soft contact lenses were shown to reduce accommodative demand at near distance tasks on digital screens, which in turn reduced eye fatigue [45]. Given the similar multifocal setup of the DIMS lenses, the added power in the peripheral zones, which leads to the desired main effect of a retardation of elongational axial length, at the same time may be able to reduce the accommodative muscle stress, resulting in an age-independent reduction in eye fatigue.

Another possibility for the reduced eye strain may lie in the fact that visual processing of information and ongoing attentional processes are impacted by the optical setup of the DIMS lenses. In particular, given that information beyond the central 13 degrees is blurred to some degree, the amount of information that needs to be processed (peripherally) is reduced compared to SV lenses. In this context, it would be interesting to test attentional processes more explicitly with DIMS and SV lenses, similar to the study of De Lestrange-Anginieur et al. [46], who investigated the complex interaction between attentional processes in the presence of blurred versus clear images in various attentional states in a spatial cueing paradigm.

Concerning the effect size of the fatigue changes, we find that the DIMS lenses decrease subjective eye fatigue by 26% in the Adolescents group and by 18% in the Adults group. This result also was confirmed in post-experiment debriefing in which participants reported that the DIMS lenses yielded a subjectively better wearing comfort and enabled them to "concentrate better" during the demanding visual search task. The ameliorating effects of the DIMS lenses also were "immediate" when comparing the first trials of the DIMS and SV conditions (Fig 3) and long-lasting with continued differences between the two lens types throughout the experiment. DIMS lenses also did not result in any adverse effects on either accuracy or response time in either group.

Although the overall difference in eye strain reduction seemed to be larger for the Adolescent group, we found no statistical evidence for differential effects of the lenses on the two groups with our sample size. We did find, however, differences in performance with evidence of significantly increased task difficulty (i.e., lower accuracy) for the younger group. Interestingly, other studies comparing children and adults in a different task have found *lower* levels of eye strain in children during video game playing in 3D viewing, for example [42]—an effect we did not see here. Future studies could extend the paradigm of DIMS lens testing also to other tasks (such as video game play).

In this context, it is worthwhile to briefly discuss potential issues resulting from the fact that our main dependent variable was based on subjective reports of eye strain. One such issue is that instructional or implicit biases may have led participants to lean towards giving DIMS lenses "better" scores, as it was not possible to "blind" participants to the type of lens worn during the testing due to the noticeably different visual input. While it is never possible to fully exclude biases, our study tried to minimize these as much as possible: participants were only informed that the lenses would potentially be beneficial in reducing myopia and that we would be comparing the effectiveness of the lenses compared to standard lenses in a behavioral task, following an acclimatization phase. The main variable of eye fatigue was *not* highlighted in the instructions to be of interest. Additionally, the initial acclimatization period–if it had an effect on ratings–could even result in a rating bias towards *more* fatiguing, as participants may have viewed the DIMS lenses as needing a long period of time to get accustomed to. Nonetheless, follow-up experiments with additional measures (such as, for example, blink rate) or even accommodation manipulation through paralysis will need to be done to provide further data on the observed effect–the ongoing COVID-19 pandemic prevented us from using such more involved experimental measures for the present study.

We also found—not surprisingly—that Easy images yielded overall lower levels of eye strain compared to Difficult images. Various research has shown that task difficulty is related to eye strain. The difficulty of the task, for example, can affect the glare response [47], which in turn can cause eye fatigue [48]. Similarly, the blink ratio is higher in straining tasks [49, 50], which constitutes another potential factor for eye fatigue. All of these factors may have played a role in the observed effect of difficulty—nonetheless, the effect of the DIMS lenses was present for both levels of difficulty and both age groups, indicating that the strain relief is a robust, general effect.

In summary, to our knowledge this is the first study to investigate the effects of DIMS lenses on eye strain via a demanding search task. We found robust decreases of eye strain for these corrective lenses, which were observed for both a younger and an older participant group, albeit at a lower level for the latter group. Future studies will extend the types of tasks and also look at potential changes in neural processes accompanying the strain reduction.

## Author Contributions

**Conceptualization:** Hyeongsuk Ryu, Christian Wallraven.

**Data curation:** Hyeongsuk Ryu.

**Formal analysis:** Hyeongsuk Ryu.

**Project administration:** Christian Wallraven.

**Writing – original draft:** Hyeongsuk Ryu.

**Writing – review & editing:** Uijong Ju, Christian Wallraven.

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
