## [Decision Letter · Decision Letter 0]

25 Jun 2021

PONE-D-21-16890

Myopia-correcting lenses decrease eye fatigue in a visual search task for both adolescents and adults

PLOS ONE

Dear Dr. Wallraven,

Thank you for submitting your manuscript to PLOS ONE. After careful consideration, we feel that it has merit but does not fully meet PLOS ONE’s publication criteria as it currently stands. Therefore, we invite you to submit a revised version of the manuscript that addresses the points raised during the review process.

We look forward to receiving your revised manuscript.

Kind regards,

Xingjun Fan, PhD

Academic Editor

PLOS ONE

Journal Requirements:

2.  Please change "female” or "male" to "woman” or "man" as appropriate, when used as a noun (see for instance https://apastyle.apa.org/style-grammar-guidelines/bias-free-language/gender).""

3. Please provide additional details regarding participant consent for minors. Specifically, in the Methods section, please state whether you obtained consent from parents or guardians. If the need for consent was waived by the ethics committee, please include this information.

Reviewers' comments:

Reviewer's Responses to Questions

**Comments to the Author**

1. Is the manuscript technically sound, and do the data support the conclusions?

Reviewer #1: Yes

Reviewer #2: No

2. Has the statistical analysis been performed appropriately and rigorously? 

Reviewer #1: Yes

Reviewer #2: No

3. Have the authors made all data underlying the findings in their manuscript fully available?

Reviewer #1: Yes

Reviewer #2: Yes

4. Is the manuscript presented in an intelligible fashion and written in standard English?

Reviewer #1: Yes

Reviewer #2: Yes

5. Review Comments to the Author

Reviewer #1: General comments:

Overall, the question and finding of this study are interesting. The descriptions are clear and organized. I suggest however the authors add an image of showing the difference between the stimuli viewed with the DMIS and the SV lens.

Specific comments:

Introduction section:

- It would be worth to add some background to introduce to the reader the different methods available to assess "eye fatigue" and explain the choice of the method presented.

Participants section:

-Was accommodation paralyzed during the test?

-Any exclusion criteria?

Section lens, stimuli:

- Providing information about the field of view corresponding to clear and blurred view of the DIMS for the image tested on the display would be helpful for the reader.

Stimuli section:

- I would suggest that the author present a representative image viewed with the SV and DIMS lenses to help the reader understanding the visual differences that may underlie the result shown in the study.

-Were they any criteria used to determine the easy from the difficult picture?

-What was the rational in the choice of the distance of the display. Since it involved a moderate effort of accommodation, it could have affected the effect of the DIMS lens.

-I think you should discuss the potential influence of the field of view in the difference observed between the two lens design. The DIMS is likely to have resulted in a smaller clear field of view, thus reducing the amount of details to be processed for detecting Waldo. In this respect, I think it would be worthwhile connecting these results with a possible influence of blur on attention (see for instance, E De Lestrange-Anginieur et al, 2021).

Section Eye fatigue:

- Please add the standard deviation corresponding to the mean where relevant.

For the effect of trials, please add the duration of the trials and test to clarify the influence of the time duration on the variation across trials.

Section Discussion:

In the post-experiment debriefing were there any specific questionnaire?

Reviewer #2: This study determined the impact of a myopic correcting lens (DIMS, Hoya corporation) in reducing visual fatigue during a visual search task, relative to regular myopic spectacles, in otherwise visually normal adults and adolescents. Participants played the Where’s Waldo game as a search paradigm with their spherocylindrical refractive error corrected using either the DIMS lens or regular spectacles. The outcomes of the search task were determined in terms of the response accuracy, response time and a subjective impression of fatigue on a five-point rating scale. The response accuracy and time did not vary significantly between the two lenses but the fatigue factor was significantly smaller in the DIMS than in regular spectacles. These reason for why result might have been obtained in speculated in the discussion section.

The manuscript is written well with limited grammatical errors. The scientific content of the manuscript is also relevant to the current context of the myopic epidemic and increased work-from-home activities owing to the COVID-19 pandemic. The following issues however need to be addressed for a complete understanding of the manuscript.

1. The hypothesis for why the DIMS design should reduce the fatigue in a visual search task and why should this be different between adults and adolescents must be made clearer. Currently, the manuscript leans on fatigue reduction with other aspheric lens designs as motivation for the present one. This is weak and does not speak directly about the DIMS design and its impact on visual fatigue.

2. Along the same lines as point #1, the results of the study only show a reduction in fatigue with the DIMS design, relative to regular spectacles, without any investigation of why this might have occurred. The discussion is all but speculations about the results and, honestly, this is rather disappointing reading it without gaining any scientific insights into why this might have occurred. At the very least, the authors should be testing their accommodation hypothesis and provide some results for or against this hypothesis in this study. The present results, in my mind, is good starting point for the investigation from only a very preliminary one and far from being complete.

3. The methods section lacks some important information. Were the subjects aware of which lens they were wearing while doing the visual search task. I would assume they were, for they have gotten used to this lens for 2 weeks prior to the start of the study. Could this prior experience have unduly influenced the results in favour of the DIMS lens? Could the investigators inadvertently have sent across a message to the participants about this lens being more “comfortable” and “fatigue resistant” relative to regular spectacles? This is a significant challenge, in my mind, with only basing the entire study on a set of subjective data that may have been influenced by all kinds of biases, without providing any additional support for their findings. All this is all the more worrying because HOYA corporation seems to have sponsored these lenses and there may be conflicts of interest in the results turning out the way they did.

4. Other information that are not available in the methods section include a clear definition of what increasing task difficulty was and how was this determined, clear definition of accuracy and response time. These are important to interpret the results shown in Tables 1 – 3.

5. Some specific points also need to be fixed:

a. The first two paragraphs of the introduction section are redundant for the purposes of this study. This can be tightened significantly into a single paragraph.

b. Page 4: No need to define OD and OS. Given the broad nature of readership of PLoS One, replace OD and OS with Right eye and left eye, respectively.

c. Page 6: Replace the word “uncomfortableness” with “discomfort”.

d. Page 6: Were the grids in the image shown to the subjects during the testing? If so, please make it explicit in the methods section.

e. Page 6: It is clear that the spectacle viewing was randomized across subjects, but were the difficulty level of the visual search task also randomized? If so, please mention it explicitly.

f. Table 1: This table is odd in that data from both types of spectacles are combined here. This should show the data of DIMS and regular spectacles separately for both participant groups.

g. Were the data normally distributed? How was this confirmed?

h. Were the ANOVA’s that were used repeated-measures ANOVA. If not, the results should be re-analyzed using repeated measures three-factor ANOVA’s.

6. PLOS authors have the option to publish the peer review history of their article (what does this mean?). If published, this will include your full peer review and any attached files.

Reviewer #1: No

Reviewer #2: No

---

## [Author Response · Author response to Decision Letter 0]

16 Sep 2021

Reviewer #1

General comments

Q1. Overall, the question and finding of this study are interesting. The descriptions are clear and organized. I suggest however the authors add an image of showing the difference between the stimuli viewed with the DMIS and the SV lens.

A1. Thank you. Pictures have been added to Figure 1 to illustrate the effects of the DIMS lenses.

Specific comments:

Introduction section:

Q2. It would be worth to add some background to introduce to the reader the different methods available to assess "eye fatigue" and explain the choice of the method presented.

A2. In the current pandemic situation, we were limited in terms of measures that would be compatible with regulations. We have added the following paragraph citing additional work to the Methods section [as this is where we introduce our questionnaire in more detail]:

“In the context of measuring eye fatigue, self-report questionnaires, critical flicker fusion frequency (CFF) [32], and blink rate measurements [41] have mainly been used as measures. Due to regulatory limitations in the ongoing COVID-19 pandemic, we were not able to employ eye-tracking with our participants, so that we decided to focus on self-report-based questionnaires in the present study [23, 39, 42]. This fatigue assessment was done across five levels, each of which was explained to participants beforehand: 1. None to very little: You did not feel any strain when finding Wally.; 2. A little: You felt a little tired as you searched for Wally; 3. Moderate: You felt tired as you searched for Wally, but it can be sustain; 4. Considerable: You feel stressed from the search task, your eyes or the area around it may feel itchy (especially the canthus), you may have had to yawn from time to time.; 5. Very Much: You found it very difficult to concentrate on the search task as your eyes felt very tired, itching or even pressure in the eyes may have occurred, you may have needed to yawn. The questionnaire was designed based on similar layouts used in previous studies [43]..”

Participants section:

Q3. Was accommodation paralyzed during the test? 

A3: No, due to medical and ethical restrictions, participants’ accommodation was not paralyzed. Please see below.

Q4. Any exclusion criteria?

A4. Exclusion criteria were determined by the constraints of the lens prescription range and were added to the manuscript in the Participants section:

 “In our selection of participants, people were excluded (1) if they wore prismatic glasses (due to an officially diagnosed strabismus), (2) if vision correction above the default dispensing range of DIMS lenses (spherical: < -10Diopter, cylindrical: <-4.00Diopter) would be required.”

Section lens:

Q5. Providing information about the field of view corresponding to clear and blurred view of the DIMS for the image tested on the display would be helpful for the reader.

A5. We agree that this should have been presented better. The corresponding numbers were added to the manuscript: 

“The different optical zones result in a blurred region subtending roughly 69˚ x 57˚ visual angle (horizontal x vertical) that excludes a central 13˚ non-blurred region.”

Stimuli section:

Q6. I would suggest that the author present a representative image viewed with the SV and DIMS lenses to help the reader understanding the visual differences that may underlie the result shown in the study.

A6. Due to licensing issues, we are not able at the present time to show an original picture for reproduction. Because of this, we have updated Figure 1 with an example of the visual effects of the DIMS lenses.

Q7. Were they any criteria used to determine the easy from the difficult picture?

A7. We agree that this point should have been explained better and we apologize for the lack of clarity here. We have updated the relevant paragraph in the Methods section to read as follows:

“A first sample of 24 pictures was obtained from the “Where is Wally?” books. In order to make the search task better to analyze, we subdivided each of the images into a regular 7x7 grid indexed by letters (‘a - g’) and numbers (‘1 – 7’). The 7x7 grid was overlaid on the stimulus images and shown during all experiments. The task for participants was to view the image and to indicate which of the target grid squares contained Wally. Based on initial inspection, we grouped the images into two different difficulty levels (Easy and Difficult) and showed these to a group of N = 10 pilot participants for validation. The number of distinct human characters in one grid square was adopted as a reference for the difficulty level decision (Easy: Min_average: 3.1ea (SD = 2.03), Max_average: 9.3ea (SD = 2.00), an example of this category is the image entitled ‘FVN and games in ancient Rome’. Difficult: Min_average: 6.6ea (SD = 3.89), Max_average: 15.5ea (SD = 6.79), an example of this category is the image entitled ‘The Mighty Fruit Fight’ - the difference in this metric was significant for both minimum and maximum values (paired t-test, Min: t(18)=-2.52, p=.021, Max: t(18)=-2.77, p=.013)).”

Q8. What was the rational in the choice of the distance of the display. Since it involved a moderate effort of accommodation, it could have affected the effect of the DIMS lens.

A8. The distance to the monitor for this task was set to 60cm – this was set as being within standard viewing distances in office environments (see Rempel, David, et al. "The effects of visual display distance on eye accommodation, head posture, and vision and neck symptoms." Human factors 49.5 (2007): 830-838.) – given that distance to the screen, however, was the same for both lens conditions, we do not think that accommodation would differentially affect wearers in both conditions.

Q9. I think you should discuss the potential influence of the field of view in the difference observed between the two lens design. The DIMS is likely to have resulted in a smaller clear field of view, thus reducing the amount of details to be processed for detecting Waldo. In this respect, I think it would be worthwhile connecting these results with a possible influence of blur on attention (see for instance, E De Lestrange-Anginieur et al, 2021).

A9. This is, indeed, a very interesting point, thank you. We have added the following paragraph to the Discussion:

“Another possibility for the reduced eye strain may lie in the fact that visual processing of information and ongoing attentional processes are impacted by the optical setup of the DIMS lenses. In particular, given that information beyond the central 13 degrees is blurred to some degree, the amount of information that needs to be processed (peripherally) is reduced compared to SV lenses. In this context, it would be interesting to test attentional processes more explicitly with DIMS and SV lenses, similar to the study of De Lestrange-Anginieur et al. [46] who investigated the complex interaction between attentional processes in the presence of blurred versus clear images in various attentional states in a spatial cueing paradigm.”

Section Eye fatigue:

Q10. Please add the standard deviation corresponding to the mean where relevant.

For the effect of trials, please add the duration of the trials and test to clarify the influence of the time duration on the variation across trials.

A10. We have added standard deviations to the results where relevant. In addition, we have added a section on the effect of trials as follows:

“Effect of Trials

Finally, we take a look at how the measures of Response Time and Eye Fatigue developed over the course of the experiment. For this, we plot both measures as a function of trial (1 to 10) and lens type (SV vs DIMS) in Figure 3. Both measures show a slight upward trend for both DIMS and SV lenses as the number of trials increases, indicating increasing fatigue levels. The two measures are also correlated significantly, as indicated by a Spearman correlation value of r=.377 (p<.001), showing that increased eye fatigue also goes along with increased response times. Importantly, however, levels for DIMS lenses stay continuously lower than those for SV lenses across all trials, whereas response times are similar across lens types.”

Reviewer #2: The manuscript is written well with limited grammatical errors. The scientific content of the manuscript is also relevant to the current context of the myopic epidemic and increased work-from-home activities owing to the COVID-19 pandemic. The following issues however need to be addressed for a complete understanding of the manuscript.

Q1. The hypothesis for why the DIMS design should reduce the fatigue in a visual search task and why should this be different between adults and adolescents must be made clearer. Currently, the manuscript leans on fatigue reduction with other aspheric lens designs as motivation for the present one. This is weak and does not speak directly about the DIMS design and its impact on visual fatigue.

A1. Thank you for this suggestion. We have updated the introduction to more directly address the issue of DIMS in terms of eye fatigue with another background study, as well as our rationale for comparing adolescents with adults in two paragraphs as follows.

“[…] As the DIMS lenses change the relative composition of focal and peripheral parts of the visual field, it is important to also investigate whether they would lead to changes in eye strain. A recent study on the DIMS lenses with two cohorts of children and adults traced wearing comfort in both groups after one week, comparing single vision (SV) lenses with DIMS lenses [33] with a one-time survey. The authors found no clear differences in the children group on measures including eye strain, nausea, or dizziness, but showed that adults tended to feel more nausea and dizziness for the DIMS compared to the SV lenses. For both groups, overall acceptance of the DIMS lenses given their potential for myopia control after the initial acclimatization phase was high.

The aforementioned study focused on evaluating the wearing comfort of myopia-control lenses with a one-time subjective evaluation. Our main aim in this study, however, was to investigate wearing comfort of such lenses for this first time in a more fine-grained manner during a demanding visual task, hence gathering more data on how SV and DIMS lenses may differentially impact perceived eye strain. For this purpose, the study by Lin et al. [32] demonstrates an interesting approach, in which a well-researched perceptual task or paradigm (like flicker fusion) is employed as a vehicle for studying factors related to eye strain and visual performance. In this manuscript, we take a similar route, using a paradigm taken from the visual search literature to investigate potential effects of DIMS lenses. Specifically, here we focus on the “Where’s Waldo” (also called “Finding Wally”) visual search puzzles. Visual search is a core perceptual task, involving the search for a particular target among a (usually complex) background [34, 35]. The “Where’s Waldo” task itself is one example of a visual search task, which has been used in several previous studies: Port et al. [36], for example, used the task to measure saccade properties across the lifespan and Sahraian et al. [37] showed that training with this task also improved target detection in radiological images. In this context, Casco et al. [38] showed that visual search performance critically depends on age, and, furthermore, work by Solimini et al. [39] indicates that eye fatigue during game play also is different by age. Given the additional, wide-ranging differences in developing and adult eyes [40] and the potential differences in sensitivity to the DIMS lenses in adults and children [33], it will therefore be important to also test how such lenses may affect adults versus adolescents differently.”

Q2. Along the same lines as point #1, the results of the study only show a reduction in fatigue with the DIMS design, relative to regular spectacles, without any investigation of why this might have occurred. The discussion is all but speculations about the results and, honestly, this is rather disappointing reading it without gaining any scientific insights into why this might have occurred. At the very least, the authors should be testing their accommodation hypothesis and provide some results for or against this hypothesis in this study. The present results, in my mind, is good starting point for the investigation from only a very preliminary one and far from being complete.

A2. Whereas we wholeheartedly agree with the reviewer’s sentiment that our research is not complete, we still think that our experimental results are novel and of interest to readers, as they are the first to find and measure a clear reduction in eye fatigue in wearers of DIMS lenses.

We mention two reasons in the discussion now: first, the additional accommodation processes that are driven by the DIMS lens design may reduce muscle fatigue: second, the peripheral blur may have reduced the load in information or attentional processing. A more stringent validation for the first point could be obtained, for example, by paralyzing accommodation in the participants and repeating the experimental procedure – our present medical and ethical regulations in the context of the ongoing COVID-10 pandemic, however, unfortunately placed restrictions on experimental procedures. We are currently setting up experiments for a second line of studies investigating the second hypothesis of attentional modulations. 

For the first point, please also see our answer to Q3. Concerning the second point, we have updated the discussion with the following paragraph:

“Another possibility for the reduced eye strain may lie in the fact that visual processing of information and ongoing attentional processes are impacted by the optical setup of the DIMS lenses. In particular, given that information beyond the central 13 degrees is blurred to some degree, the amount of information that needs to be processed (peripherally) is reduced compared to SV lenses. In this context, it would be interesting to test attentional processes more explicitly with DIMS and SV lenses, similar to the study of De Lestrange-Anginieur et al. [46], who investigated the complex interaction between attentional processes in the presence of blurred versus clear images in various attentional states in a spatial cueing paradigm.”

Q3. The methods section lacks some important information. Were the subjects aware of which lens they were wearing while doing the visual search task. I would assume they were, for they have gotten used to this lens for 2 weeks prior to the start of the study. Could this prior experience have unduly influenced the results in favour of the DIMS lens? Could the investigators inadvertently have sent across a message to the participants about this lens being more “comfortable” and “fatigue resistant” relative to regular spectacles? This is a significant challenge, in my mind, with only basing the entire study on a set of subjective data that may have been influenced by all kinds of biases, without providing any additional support for their findings. All this is all the more worrying because HOYA corporation seems to have sponsored these lenses and there may be conflicts of interest in the results turning out the way they did.

A3. We agree with the reviewer that implicit biases or instructional effects would potentially be detrimental to our study. We did try to minimize such biases by avoiding to highlight our main variable of interest in the instructions. Given the lengthy acclimatization phase, one may also assume that participants would be inclined towards more eye strain for the DIMS lenses. We have updated this issue in the discussion, also highlighting the need for further, more detailed experiments with other measures in this context.

The following paragraph was added to the Discussion:

“In this context, it is worthwhile to briefly discuss potential issues resulting from the fact that our main dependent variable was based on subjective reports of eye strain. One such issue is that instructional or implicit biases may have led participants to lean towards giving DIMS lenses “better” scores, as it was not possible to “blind” participants to the type of lens worn during the testing due to the noticeably different visual input. While it is never possible to fully exclude biases, our study tried to minimize these as much as possible: participants were only informed that the lenses would potentially be beneficial in reducing myopia and that we would be comparing the effectiveness of the lenses compared to standard lenses in a behavioral task, following an acclimatization phase. The main variable of eye fatigue was not highlighted in the instructions to be of interest. Additionally, the initial acclimatization period of up to two weeks – if it had an effect on ratings – could even result in a rating bias towards more fatiguing, as participants may have viewed the DIMS lenses as needing a long period of time to get accustomed to. Nonetheless, follow-up experiments with additional measures (such as, for example, blink rate) or even accommodation manipulation through paralysis will need to be done to provide further data on the observed effect – the ongoing COVID-19 pandemic prevented us from using such more involved experimental measures for the present study.”

Finally, we would like to again stress that the HOYA corporation had no influence on the design, analysis, or interpretation of the result as affirmed in the manuscript under Acknowledgments. 

Q4. Other information that are not available in the methods section include a clear definition of what increasing task difficulty was and how was this determined,/ clear definition of accuracy and response time. These are important to interpret the results shown in Tables 1 – 3.

A4. We agree with the reviewer that our description of task difficulty was unclear and have updated the methods section as follows:

“A first sample of 24 pictures was obtained from the “Where is Wally?” books. In order to make the search task better to analyze, we subdivided each of the images into a regular 7x7 grid indexed by letters (‘a - g’) and numbers (‘1 – 7’). The 7x7 grid was overlaid on the stimulus images and shown during all experiments. The task for participants was to view the image and to indicate which of the target grid squares contained Wally. Based on initial inspection, we grouped the images into two different difficulty levels (Easy and Difficult) and showed these to a group of N = 10 pilot participants for validation. The number of distinct human characters in one grid square was adopted as a reference for the difficulty level decision (Easy: Min_average: 3.1ea (SD = 2.03), Max_average: 9.3ea (SD = 2.00), an example of this category is the image entitled ‘FVN and games in ancient Rome’. Difficult: Min_average: 6.6ea (SD = 3.89), Max_average: 15.5ea (SD = 6.79), an example of this category is the image entitled ‘The Mighty Fruit Fight’ - the difference in this metric was significant for both minimum and maximum values (paired t-test, Min: t(18)=-2.52, p=.021, Max: t(18)=-2.77, p=.013)).”

Q5. Some specific points also need to be fixed:

a. The first two paragraphs of the introduction section are redundant for the purposes of this study. This can be tightened significantly into a single paragraph.

A5a. We tried to tighten this into one paragraph, but ended up with a very compressed introduction. We therefore kept the original version in this revision, as we ultimately felt that it would perhaps be best for the general readership of PLoS One to use the broad issue of increasing myopia as a first hook, and clinical interventions for treatment of myopia as a more specific dive into our topic.

b. Page 4: No need to define OD and OS. Given the broad nature of readership of PLoS One, replace OD and OS with Right eye and left eye, respectively.

A5b. We agree and this is corrected.

c. Page 6: Replace the word “uncomfortableness” with “discomfort”. 

A5c. This is corrected.

d. Page 6: Were the grids in the image shown to the subjects during the testing? If so, please make it explicit in the methods section.

A5d. Yes, this was updated in the Methods section [see A4].

e. Page 6: It is clear that the spectacle viewing was randomized across subjects, but were the difficulty level of the visual search task also randomized? If so, please mention it explicitly.

A5e. Yes, again this was updated in the Stimuli section: “Images were randomized within each set and sets were randomized across participants.”

f. Table 1: This table is odd in that data from both types of spectacles are combined here. This should show the data of DIMS and regular spectacles separately for both participant groups.

A5f. Thank you for this suggestion. We now report data for both groups separately.

g. Were the data normally distributed? How was this confirmed?

A5g. Although ANOVAs in general are reasonably robust to violations of normality in many cases, we thank the reviewer for this pointer. In light of this, we have taken additional measures and updated the statistics accordingly – although individual statistical values changed somewhat, none of the significant effects changed. 

The following parts were added to Data analysis: 

“Prior to the main testing, Shapiro-Wilk tests and additional Skewness and Kurtosis checks were conducted to confirm normality, followed by transformations if necessary. Mixed ANOVAs were then used for analyses of main effects and interactions using SPSS”

In Results:

“Given the large number of trials (n=840), Shapiro-Wilk tests indicated violation of normality for all variables, however, skewness and kurtosis were within ±2 for Accuracy and Eye Fatigue. The variable of Response Time showed a highly skewed, left-tailed histogram, and therefore was log-transformed before analysis with the mixed ANOVA.“

h. Were the ANOVA’s that were used repeated-measures ANOVA. If not, the results should be re-analyzed using repeated measures three-factor ANOVA’s.

A5h. See above. Data were analyzed using mixed ANOVAs that tested both within- and between-participant effects.

---

## [Decision Letter · Decision Letter 1]

28 Sep 2021

Myopia-correcting lenses decrease eye fatigue in a visual search task for both adolescents and adults

PONE-D-21-16890R1

Dear Dr. Christian Wallraven,

We’re pleased to inform you that your manuscript has been judged scientifically suitable for publication and will be formally accepted for publication once it meets all outstanding technical requirements.

Kind regards,

Xingjun Fan, PhD

Academic Editor

PLOS ONE

Additional Editor Comments (optional):

All comments have been addressed.

Reviewers' comments:

Reviewer's Responses to Questions

**Comments to the Author**

1. If the authors have adequately addressed your comments raised in a previous round of review and you feel that this manuscript is now acceptable for publication, you may indicate that here to bypass the “Comments to the Author” section, enter your conflict of interest statement in the “Confidential to Editor” section, and submit your "Accept" recommendation.

Reviewer #1: All comments have been addressed

2. Is the manuscript technically sound, and do the data support the conclusions?

Reviewer #1: Yes

3. Has the statistical analysis been performed appropriately and rigorously? 

Reviewer #1: Yes

4. Have the authors made all data underlying the findings in their manuscript fully available?

Reviewer #1: Yes

5. Is the manuscript presented in an intelligible fashion and written in standard English?

Reviewer #1: Yes

6. Review Comments to the Author

Reviewer #1: 

7. PLOS authors have the option to publish the peer review history of their article (what does this mean?). If published, this will include your full peer review and any attached files.

Reviewer #1: No

---

## [Editor Report · Acceptance letter]

4 Oct 2021

PONE-D-21-16890R1 

Myopia-correcting lenses decrease eye fatigue in a visual search task for both adolescents and adults 

Dear Dr. Wallraven:

I'm pleased to inform you that your manuscript has been deemed suitable for publication in PLOS ONE. Congratulations! Your manuscript is now with our production department. 

Kind regards, 

on behalf of

Dr. Xingjun Fan 

Academic Editor

PLOS ONE